behaviour

distress calls, communication, bats, sex differences, vocalizations

**Authors for correspondence:**
Eugenia González-Palomares
e-mail: gonzalezpalomares@bio.uni-frankfurt.de
Julio C. Hechavarria
e-mail: hechavarria@bio.uni-frankfurt.de

# Male *Carollia perspicillata* bats call more than females in a distressful context

Eugenia González-Palomares, Luciana López-Jury, Johannes Wetekam, Ava Kiai, Francisco García-Rosales and Julio C. Hechavarria

Institute for Cell Biology and Neuroscience, Goethe University, 60438 Frankfurt am Main, Germany

EG-P, 0000-0003-3086-7601; LL-J, 0000-0002-9384-2586; JW, 0000-0003-1919-3140; AK, 0000-0001-6996-0868; FG-R, 0000-0001-5576-2967; JCH, 0000-0001-9277-2339

Distress calls are a vocalization type widespread across the animal kingdom, emitted when the animals are under duress, e.g. when captured by a predator. Here, we report on an observation we came across serendipitously while recording distress calls from the bat species *Carollia perspicillata*, i.e. the existence of sex difference in the distress calling behaviour of this species. We show that in *C. perspicillata* bats, males are more likely to produce distress vocalizations than females when hand-held. Male bats call more, their calls are louder, harsher (faster amplitude modulated) and cover lower carrier frequencies than female vocalizations. We discuss our results within a framework of potential hormonal, neurobiological and behavioural differences that could explain our findings, and open multiple paths to continue the study of sex-related differences in vocal behaviour in bats.

## 1. Introduction

Vocal communication is essential as a means of interaction among living beings. One example of this is distress calls, a type of vocalization widespread across the animal kingdom, present in humans [1,2], non-human primates [3], lizards [4], crocodilians [5], frogs [6], rats [7], birds [8] and bats [9,10], among other species. Distress calls are uttered when the animals find themselves in a situation of utmost danger, such as just before or during the capture by a predator [11–13]. These calls are typically produced at fast rates, cover a broadband frequency spectrum, travel long distances, and their source is easy to locate [10,11].

Although distress calling behaviours have been well researched in the last decades, the reasons why animals produce

these sounds are still not fully understood. Five hypotheses have been proposed which are not mutually exclusive [8,11]: (i) warning kin, increasing the caller's inclusive fitness (number of offspring) by helping its kin avoiding predators; (i) requesting aid from kin and other individuals (reciprocal altruists) to help the caller escape; (iii) attract an audience that will use that information to escape predation in the future; (iv) attracting another predator which could disturb the initial predator providing an opportunity to escape and (v) startling the predator into releasing the caller.

Despite all the studies on the function of distress calls, their fitness benefits are still uncertain [7–9,11,12,14–16]. Surveying the caller's social behaviour, observing its effect on the listeners, documenting sex-specific differences and distinguishing between different calls are all approaches for exploring the potential functions of distress vocalizations. Solitary bird species have been observed to emit distress calls with a lower probability than flocking or gregarious species [17,18]. Male baboons respond more to distress calls from females they have a close relationship with than to control females [19]. Playback experiments show that sympatric species can be responsive to distress calls [15,20,21]. In bats, the location where the calls are emitted (i.e. how far away from the day roost) also affects responsiveness to distress signals [22].

This paper aims to quantitatively explore an observation we came across serendipitously while recording distress calls from the bat species *Carollia perspicillata*: there seem to exist sex differences in the distress calling behaviour of this species. Specifically, in previous experiments, we noticed that male bats appear to be the most vocal sex in this species, but this qualitative observation was never researched thoroughly. The emission pattern and neuronal processing of distress calls emitted by *C. perspicillata* have been investigated in previous studies [10,23,24]. This bat species emits distress sequences temporally arranged into syllable groups (bouts) [10] and individual syllables carry amplitude fluctuations at rates of approximately 1.7 kHz that could represent an acoustic correlate of perceptual roughness [25]. None of the previous studies on distress calling in *C. perspicillata* explored possible sex differences in calling patterns. Sex-dependent calling behaviours are rare among mammals, and they are considered a useful tool for understanding the determinants of vocal communication and its evolution [26,27].

In bats, prior research has shown sex differences in distress calling behaviour in species such as *Sturnira lilium* in which males were more likely to produce distress sounds than females [14]. However, in other bat species (*Glossophaga soricina* [28] and *Rhinolophus ferrumequinum* [29]), this sex distinction in distress calling was not as clear. Although these studies have alluded to sex differences—or its absence—in bat distress calling behaviour, in-depth information about sex differences—i.e. how acoustic parameters of the calls change with sex—is presently lacking. Here, we tackled this issue by conducting a thorough quantification of distress calling behaviour in adult *C. perspicillata* bats. Sex differences were analysed in terms of number of calls, loudness, peak frequency, call duration and acoustic roughness (amplitude modulations). Our results showed that, as hypothesized, male *C. perspicillata* bats call more than females when under duress. This result is discussed in the light of possible mechanisms that make males more prone to call and the implications of the latter from an evolutionary perspective.

# 2. Results

## 2.1. Male bats vocalize more, louder and lower in frequency

We recorded distress vocalizations from 61 bats (21 females) of the species *Carollia perspicillata* while hand-holding them and carefully caressing the neck [10,14]. The recordings of the three females and males that vocalized the most are shown in figure 1a. Note the difference in the number of vocalizations between sexes. We manually discarded echolocation calls from the recordings and analysed the remaining vocalization units, i.e. distress syllables [10]. In total, 7832 syllables were analysed. Figure 1b–e shows spectrograms, oscillograms, energy envelopes and the temporal modulation spectra (TMS) of examples of distress syllables emitted by female and male bats. Spectral and temporal parameters measured in individual calls were used for comparing distress calling behaviour between sexes (see below).

In accordance with what other studies suggested [14] and with our hypothesis, male bats uttered a higher number of distress vocalizations than females (figure 2a, Wilcoxon rank-sum test, $p < 0.001$; large effect size, Cliff's $d = 0.60$). Eighteen (out of 21) females and 37 (out of 40) males uttered at least one distress call. Male vocalizations were louder (figure 2b, Wilcoxon rank-sum tests, $p < 10^{-37}$; medium effect size, Cliff's $d = 0.37$), slightly but significantly shorter (figure 2c, Wilcoxon rank-sum

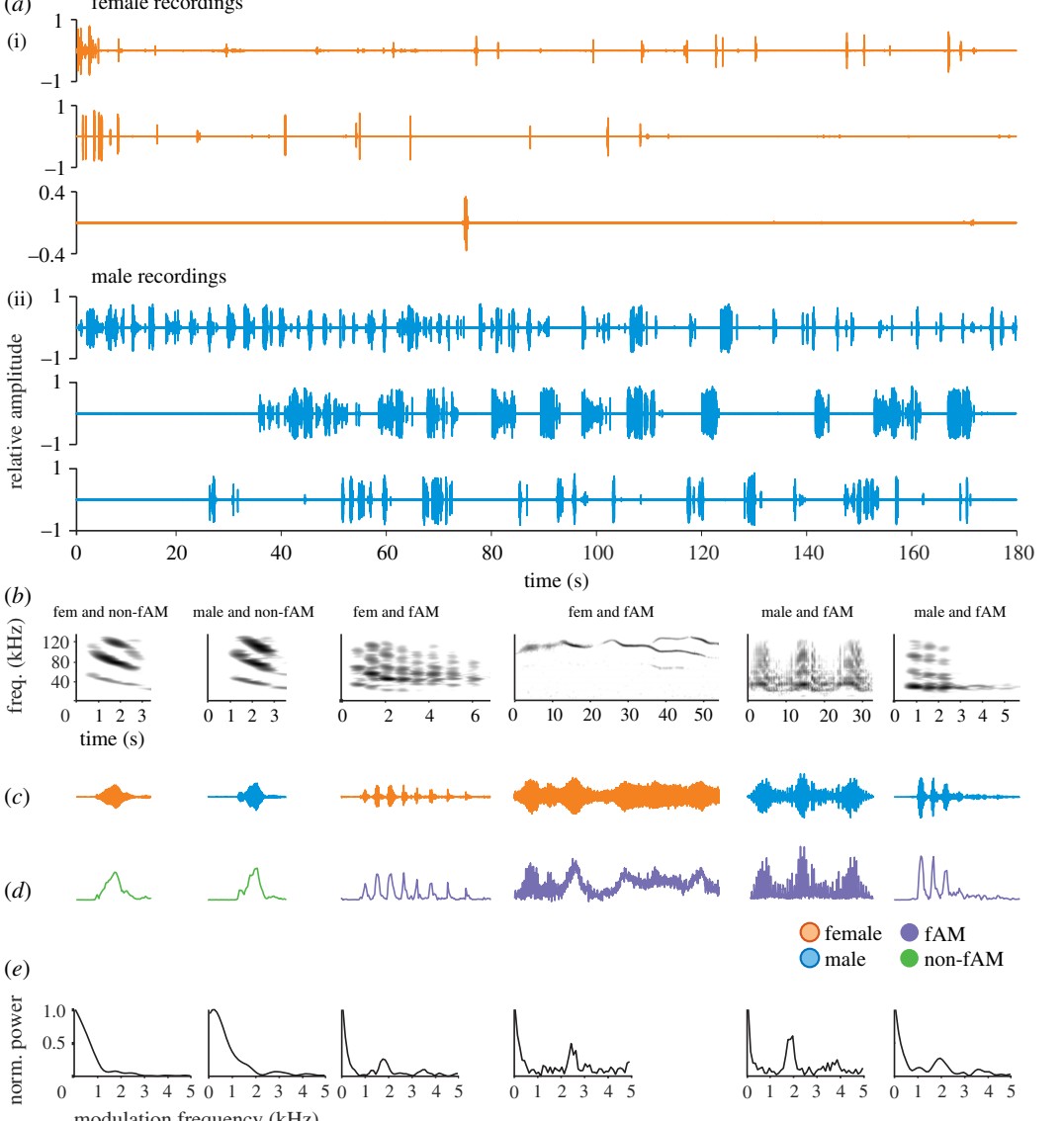

**Figure 1.** Examples of recordings and distress syllables. (a) Full recordings of the three female (i) and male (ii) bats that vocalized the most. (b–e) Examples of distress syllables of fAM (purple) and non-fAM (green) vocalizations for both females (orange) and males (blue). (b) Spectrograms, (c) time waveforms, (d) envelope and (e) temporal modulation spectra. Note the peak at approximately 1.7 kHz for fAM calls. Fem: female. fAM: fast amplitude modulated.

tests, $p < 10^{-13}$; small effect size, Cliff's $d = 0.22$), with slightly larger bandwidth (figure 2d, Wilcoxon rank-sum tests, $p < 10^{-4}$; negligible effect size, Cliff's $d = 0.12$) and had lower peak frequency than female vocalizations (figure 2e, Wilcoxon rank-sum tests, $p < 10^{-55}$; medium effect size, Cliff's $d = 0.45$). The median spectrum of all male vocalizations (figure 2f, blue line) corresponds with previously described distress calls for *C. perspicillata* [10], which reported peak frequencies at 23 kHz. On the other hand, the median spectrum for female calls showed that the frequencies with the highest power were around 75–90 kHz (figure 2f, orange line) peak frequency: $91 ± 0.1$ kHz, mean ± s.e.m. Note that these high-frequency vocalizations are different from echolocation calls (see biosonar call data in electronic supplementary material, figure S1f). Therefore, female distress calls differ from those of males in their frequency composition (see examples in figure 1b).

In females, the peak frequency of distress syllables is significantly correlated with the forearm length (figure 2g(i), $r = 0.47$, $p < 0.05$), while in males, the peak frequency tends to increase with body mass (figure 2g(ii), $r = 0.28$, $p = 0.1$). Similar analyses were conducted for echolocation calls recorded in the same animals (electronic supplementary material, figure S1) and rendered no large sex differences producing in all cases small or negligible effect sizes.

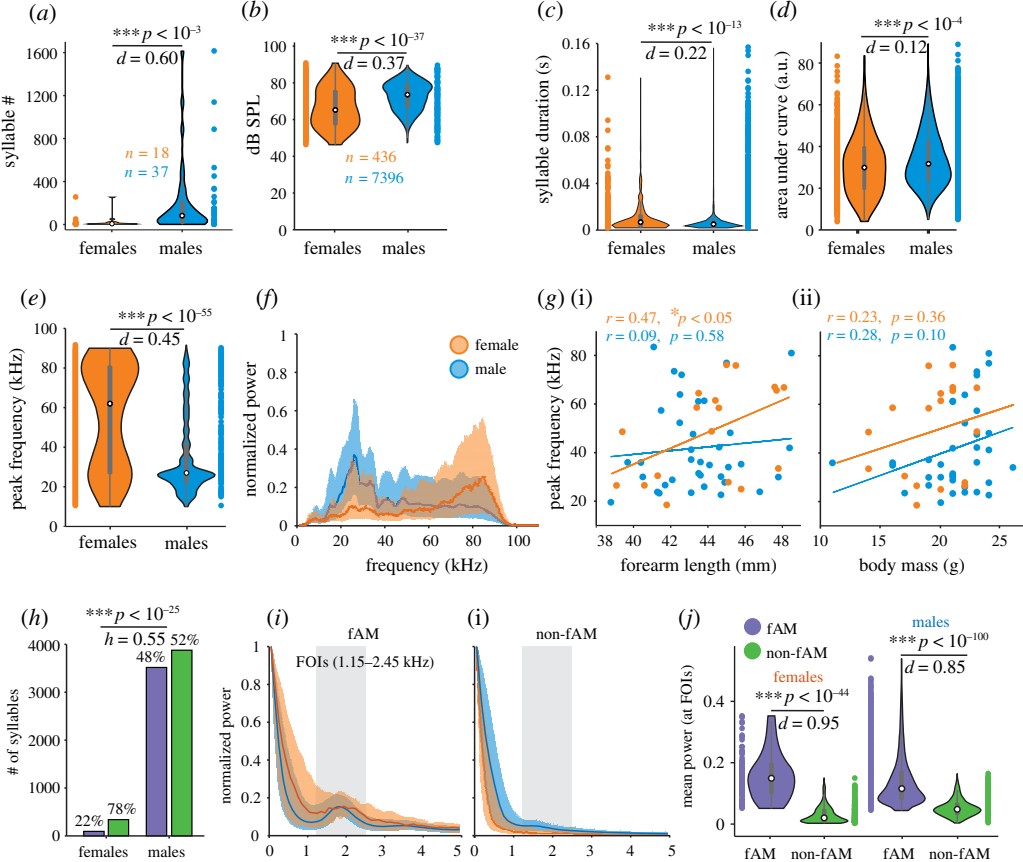

**Figure 2.** Male bats vocalize more distress calls, louder, with lower peak frequency and faster amplitude modulations than females. (*a*) Violin plot of the number of syllables per bat for females and males (Wilcoxon rank-sum test, ***$p < 0.001$). *n*: number of bats that vocalized at least one distress call. (*b*) Violin plot of the dB SPL for females and males (Wilcoxon rank-sum test, ***$p < 10^{-37}$). *n*: number of vocalizations. (*c*) Violin plots for syllable duration for females and males (Wilcoxon rank-sum test, *** $p < 10^{-13}$). (*d*) Violin plot of the area under the curve of the FFT (between 5 and 100 kHz) of the syllables as a proxy for the bandwidth for females and males (Wilcoxon rank-sum test, ***$p < 10^{-4}$). (*e*) Violin plot of the peak frequency in female and male syllables (Wilcoxon rank-sum test, ***$p < 10^{-55}$). (*f*) Median (line) and interquartile range (shaded area) of the distress calls' spectra for females (orange) and males (blue). (*g*) Scatter plots of the mean peak frequency of all the syllables per bat and the forearm length and body mass for females (orange) and males (blue). The lines represent the linear regression fit with the corresponding colour (the linear regression equations, Pearson's correlation coefficient and *p*-values are as follows: forearm length; females; $y = 3.30x − 96.65$, $r = 0.47$, $p < 0.05$; males, $y = 0.77x + 8.49$, $r = 0.09$, $p = 0.58$; body mass; females, $y = 1.63x + 17.42$, $r = 0.23$, $p = 0.36$; males, $y = 1.79x + 3.99$. $r = 0.28$, $p = 0.10$). (*h*) Histogram of the syllables classified into fAM and non-AM in both females and males, with its percentage for each sex (statistical comparison of the proportion: Chi-squared test, $p < 10^{-25}$). (*i*) Median (line) and interquartile intervals (colour shaded area) of the TMS for females (orange) and males (blue). Note the peak at the FOIs (grey-shaded area) in the fAM group and its absence in the non-fAM group. (*j*) Mean power at the FOIs of the syllables for fAM and non-fAM for each sex (Wilcoxon rank-sum test; females, $p < 10^{-44}$; males, $p < 10^{-100}$). FOI: frequency of interest. *h*: Cohen's h. *d*: Cliff's delta.

## 2.2. Male vocalizations have faster amplitude modulations

In humans, scream vocalizations (by hypothesis a type of distress call) have strong amplitude modulations between 30 and 150 Hz, corresponding to the acoustic correlate of perceptual roughness [1]. In *C. perspicillata*, periodicity values are more than 10 times faster than in humans (approx. 1.7 kHz) and have been proposed as the parameter that renders bat vocalizations rough or harsh [25]. This characteristic is common in distress calls [30,31] and makes them easier to localize [32]. Hence, we wanted to test if its presence in distress vocalizations was also sex biased.

We classified the distress syllables into fast amplitude modulated (fAM) and non-fAM vocalizations, using a binary support vector machine classification algorithm. As a training set, we used vocalizations (50 fAM and 50 non-fAM) from another dataset [25]. Note the peak at approximately 1.7 kHz for the fAM calls in the TMS (figure 1*e*). Forty-eight per cent ($n = 3518$) of the male vocalizations and 22%

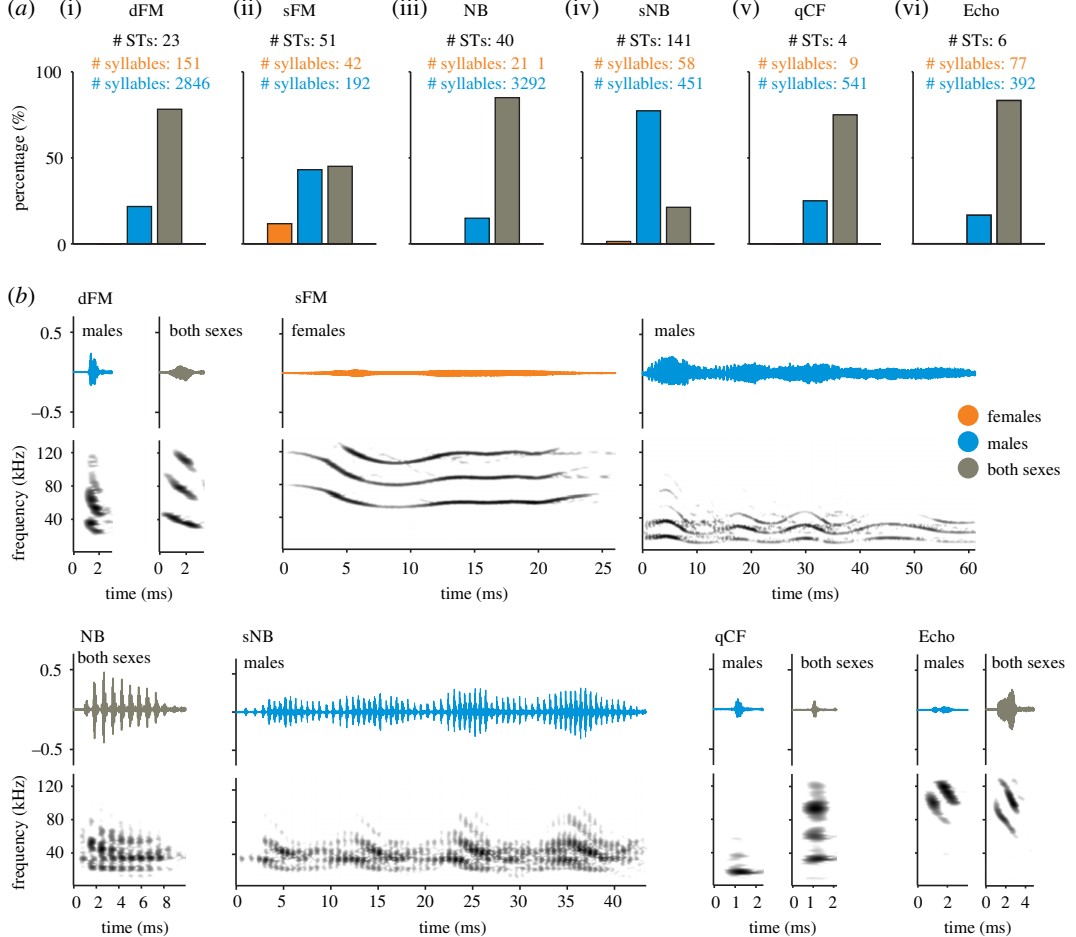

**Figure 3.** Distress syllables classified into five categories. (*a*) Histograms of the percentage of STs in each distress calls' category. Each histogram shows the percentage of STs unique to females (orange), unique to males (blue) and present in both sexes (grey) together with the total number of STs and syllables, i.e. vocalizations, for each category. dFM: down frequency modulated, sFM: sinusoidal frequency modulated, NB: noise burst, sNB: sinusoidal noise burst, qCF: quasi constant frequency, Echo: echolocation calls. (*b*) Examples of syllables of different categories, with specifications as in (*a*). The examples shown in grey belong to STs present in both females and males, although the ones shown here were uttered by males. (ST: syllable type).

($n = 95$) of the female calls were classified as fAM (figure 2*h*). These proportions are statistically different (Chi-squared test: $X_1^2 = 110.1$, $p < 10^{-25}$; medium effect size, Cohen's $h = 0.55$). The median and interquartile range of the TMS are represented in figure 2*i*. As expected, in both sexes, rough-like vocalizations (fAM) have a higher mean power at the frequencies of interest (FOIs: 1.15–2.45 kHz; Wilcoxon rank-sum test; females, $p < 10^{-44}$; males, $p < 10^{-100}$; figure 2*j*).

## 2.3. Distribution of syllable types in females and males

We studied the possibility that females and males produced different vocalization types (i.e. spectro-temporal designs) when calling in distressful situations. To that end, all syllables recorded were automatically classified into 265 syllable types (STs) using spectral cross-correlation (algorithm included in Avisoft SAS Lab Pro, see Methods). These STs were grouped into echolocation calls and five main categories of distress calls (modified from [33]) by visual inspection of the templates included in each category. The five call categories studied were downward frequency-modulated sounds (dFM), sinusoidal frequency modulated (sFM), noise burst (NB), sinusoidal noise burst (sNB) and quasi constant frequency (qCF).

We observed that male and female vocalizations are not equally distributed within the categories (figure 3*a*, Chi-squared test: $X_5^2 = 176.1$, $p < 10^{-25}$; Cramér's $V = 0.15$, medium effect size). Overall, in each category, a large number of STs were shared between females and males (figure 3). However, a large percentage of STs occurred exclusively in males. Note that this could be linked to the fact that

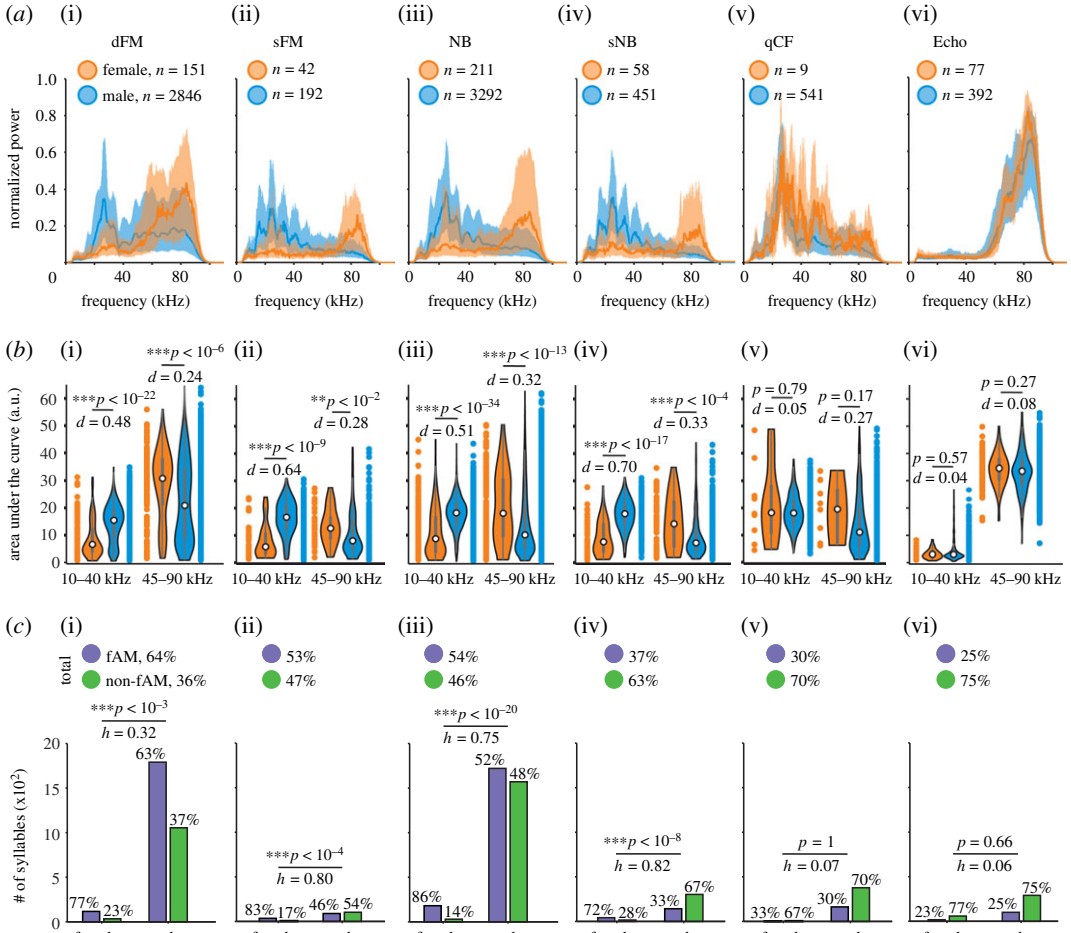

**Figure 4.** Spectral and fAM properties. (*a*) Median (line) and interquartile range (shaded area) of the distress calls' spectra for females (orange) and males (blue) and for each ST. dFM: down frequency modulated, sFM: sinusoidal frequency modulated, NB: noise burst, sNB: sinusoidal noise burst, qCF: quasi constant frequency, Echo: echolocation calls. *n*: number of vocalizations. (*b*) Violin plots and statistical comparisons (Wilcoxon rank-sum tests) between females and males of the area under the curve in the low-frequency (10–40 kHz) and high-frequency (45–90 kHz) range, for each syllable type. *$p < 0.05$. ***$p < 0.001$. *d*: Cliff's delta. (*c*) Top, percentage of syllables classified into fAM and non-AM for each ST group. Bottom, histogram of the syllables classified into fAM and non-AM in both females and males, with the percentage for each sex (statistical comparison of the proportion: Chi-squared test, ***$p < 0.001$). *h*: Cohen's h.

males called more than females (see preceding text). Templates that occurred exclusively in females were rare, but more abundant in the sFM category.

Examples of common and sex-exclusive call designs are shown in figure 3*b*. Note that automatic classification of calls based on their spectrograms is subject to many factors including call duration and frequency content. These factors could have played a role in the classification of long sounds, such as sFMs. For instance, the two sFM examples shown in figure 3*b* where sex-specific (one template was found exclusively in females, the other one only in males) but sFMs as a category occurred in both sexes, albeit with sex-specific differences in call duration, frequency composition, etc. In other words, changes in duration and pitch (but not in design) could be responsible for the female-specific templates found in the sFM category as well as for the abundant male-specific call designs observed across categories.

To quantify frequency differences between female- and male-emitted calls, we computed average spectra (for each sex) in all the categories, including echolocation calls (figure 4*a*). Note that there are fewer female vocalizations than male ones. Nonetheless, when statistically comparing the area under the curve of the spectra in the low- and high-frequency ranges (10–40 kHz and 45–90 kHz, respectively; figure 4*b*), male and female vocalizations differ statistically in all distress categories except for qCF. Additionally, in the categories dFM, sFM, NB and sNB, the proportion of fAM and non-fAM in females and males was significantly different (figure 4*c*). In echolocation calls, there was neither a sex difference in the two frequency ranges nor in the proportion of fAM/non-fAM

vocalizations. In conclusion, studying the types of vocalizations uttered by females and males indicates a large convergence in spectro-temporal design in both sexes. Thus, sex-specific differences in distress calling behaviour do not appear to be linked to the use of distinct call types in males and female bats.

# 3. Discussion

The main purpose of this article was to analyse sex differences in distress vocalizations of the bat *C. perspicillata*. We report significant sex differences in distress calling behaviour. Our main result is that male bats utter more vocalizations than females. In addition, male distress calls are louder, have lower peak frequencies, stronger amplitude modulations and are slightly shorter and broader in their frequency spectrum than female calls.

Previous studies in bats on sex differences in vocal behaviour are scarce. Carter *et al.* [14] reported that male bats of the species *Sturnira lilium* are more likely to produce distress calls than females. On the contrary, a large proportion of females and males of the species *Glossophaga soricina* emitted distress calls (65% and 50%, respectively) when handled, but no significant differences were found [28]. Male and female *Rhinolophus ferrumequinum* do not show sex-specific behaviour regarding distress calling in parameters like call rate, fundamental frequency or duration [29]. In this article, we document clear sex differences in distress calling behaviour in *C. perspicillata* bats. Although the experimental conditions here differ from the aforementioned studies, this behaviour seems to be species-specific. Future research is needed in order to establish which species from the same and other taxa show this distinction, which could help understand the purpose of distress calls and the origins of this and other sexual dimorphisms.

Sex differences in the animals' behaviour arise from differences in the combination of both genetic and hormonal factors together with environmental events during development and adulthood [34]. One possible speculation is that the distress calling dimorphisms presented here are linked to gonadal steroids at the time of testing and/or the effects of steroids during development. Another hypothesis is that the larger amount of distress calling in male *C. perspicillata* could be linked to higher levels of aggression in males. Note that these two hypotheses are not mutually exclusive, as hormonal levels have been linked to the level of aggressiveness observed in several animal species [35–37]. A third hypothesis is that this calling behaviour is learned by male bats during development. Bats are one of the few mammalian taxa that exhibit vocal learning [38,39]. We want to emphasize that the data presented in this manuscript only describes the vocal sexual dimorphism in *C. perspicillata* bats. Which—if any—of the above hypotheses explains this dimorphism in distress calling behaviour remains to be tested. Below, we discuss in more detail the internal and external factors that could potentially sculpt distress calling in bats.

## 3.1. Internal and external factors that could potentially shape distress calling behaviour in bats

There is abundant literature indicating that in many mammal species—including humans—females are often regarded as less aggressive than males [40–42]. However, in the reproduction context, females can display maternal aggression towards others, particularly towards males [43,44]. On the other hand, males generally show aggression towards both sexes [42,45]. One factor influencing aggressive behaviour is the level of testosterone [46]. For example, mice with brain-specific deletion of the androgen receptor gene show less aggression in the resident-intruder paradigm compared to control mice [47]. Similarly, rhesus monkeys with higher testosterone levels manifest more dominant and aggressive behaviours [36]. Japanese quail chicks modulate the isolation-driven distress calling behaviour in response to subcutaneous administration of testosterone [48]. If distress calling in bats is correlated with aggression and testosterone levels, then this could provide support for the 'startle the predator' hypothesis.

Other hormonal systems that are sexually dimorphic, and which are associated with the mammalian reproductive and social behaviours, are the oxytocin (OT) and arginine vasopressin (AVP) systems [49–51]. In non-human studies, these systems have been associated with sex-specific social behaviours [52], including vocal production. For example, intranasal administration of OT and AVP in female rhesus macaques resulted in an increase of aggression and 'cooing' vocalizations towards males, but not females [49]. This evidence makes OT and AVP potential candidates for explaining the sex difference in distress calling reported here.

Another possible explanation for the increased rate of distress calling in the bat species used in this study is the impact of stress. In chicks (*Gallus gallus* dom.), distress calling in social isolation contexts has been associated with stressful situations [53]. High levels of stress biomarkers, such as corticosterone and

interleukin-6, were positively correlated with the amount of distress calls [54,55]. The distress calling behaviour was also influenced by the administration of anxiolytics and antidepressants [55]. Future studies could try to characterize the relation between different stress biomarkers and distress calling in *C. perspicillata* bats of both sexes. Additional elements not discussed here could be involved in the differences in distress calling behaviour observed in adult *C. perspicillata*. These include, for example, different neurotransmitter systems and brain structures that could play a role in vocal production under duress and that could potentially be susceptible to sex differences.

## 3.2. General considerations

Behavioural actions have manifold underlying indicators, some being originators of others. Here, we have described how distress calling differs between males and females of the same bat species. It is important to mention that there exists some level of intrasexual variability, i.e. some females vocalized more than others even though as a group females vocalized less than males. In other words, all the hormonal and environmental factors mentioned above likely shape the behaviour of each bat in an individual-specific manner. It is worth noting also that other distressful contexts, such as captured by a natural predator, could have different results than have been reported here.

In addition, in bats, vocalization features vary in a continuous, rather than in a discrete manner. This makes their classification challenging. Also distress call types used by males and females differ to some degree. For instance, a higher level of fAM in the syllables uttered might signal a greater urgency [25]. Fast amplitude modulations are more prominent in male bats, and this behaviour might be affected but not determined by the molecular pathways that establish sex, as well as other environmental factors which might come into play.

Whether the higher number of uttered distress calls in males can be extrapolated to other communication calls common in both sexes is still unknown. Furthermore, male *C. perspicillata* have a richer vocal repertoire; they exclusively produce courtship and aggressive trills in this species [56]. Having a larger vocal repertoire could influence the production of other vocalization types. This might impact other areas of study in bats, such as vocal learning, e.g. in the choice of sex and/or paradigm in experimental studies. Evidence in favour of vocal learning in bats has been observed in male-specific calls [57,58] as well as in calls used by both sexes during specific lifespan periods [59–62]. However, other studies on presumably common vocalization types, such as the non-aggressive social calls in adult *Phyllostomus discolour*, have examined only males [38,39]. Intriguingly, in adult male zebra finches, brain-derived estrogens (neuroestrogens) were found to be important for learning new sounds [63]. Sex differences in vocal learning could be the manifestation of more sex-specific traits. However, since bats belong to a very diverse taxon, this should be addressed in a species-specific manner.

In conclusion, our study provides a clear report of sexual dimorphism in distress calling behaviour in adult *C. perspicillata* bats. While this paper does not tackle the mechanisms behind sex differences in vocal production under duress, it opens an important gate for future studies dealing with the external and internal factors underlying distress calling.

# 4. Material and methods

Sixty-one bats (21 females) of the species *Carollia perspicillata* were used for this study. The bats were taken from the bat colony at the Institute for Cell Biology and Neuroscience at the Goethe University in Frankfurt am Main, Germany. The experiments comply with all current German laws on animal experimentation (Regierungspräsidium Darmstadt, experimental permit # F104/57).

## 4.1. Data acquisition

The bats were taken in small groups from the colony and brought to a sound-proof booth for the individual vocalizations recording (3 min long), body mass (measured to the nearest 1 g) and forearm length measurements (measured with calipers to the nearest 0.1 mm). For the recording, the animals were hand-held with the face aiming at the microphone (Brüel & Kjaer, ¼ inch Microphone 4135, Microphone Preamplifier 2670) located approximately 0.5 m away. To encourage vocal production, the neck of the animals was softly caressed by the experimenter [10]. Sounds were digitized at 384 kHz (ADI-2 Pro, RME, 16-bit precision) and acquired by means of the BatSound software (Pettersson Elektronik, Sweden).

## 4.2. Analysis

Digitized sounds were saved in the computer for off-line analyses using Avisoft SAS Lab Pro software (v. 5.2 Avisoft Bioacoustics, Germany). The recordings were high pass filtered (cut-off: 5 kHz; eighth-order Butterworth filter). Syllables, i.e. vocalization units, were detected first using an automatic procedure that recognized all threshold crossings (4.1% of the maximum amplitude of the recording) that lasted longer than 1 ms. Second, these detections were manually revised in order to supervise the accuracy of the results. The syllables were classified into two groups, 'echolocation' and 'distress', by visual inspection of their oscillograms and spectrograms. The syllables with short duration (approx. 2 ms) and with no harmonics at low frequencies (20–30 kHz) were classified as 'echolocation' calls [64], while the rest was pooled into the 'distress' group.

All further analyses regarding gender comparisons were done in custom-written scripts in Matlab (R2019b, The MathWorks, Massachusetts, USA). The syllables were classified into fAM or non-fAM vocalizations based on their TMS using a binary support vector machine classifier. The classifier (*fitcsvm* function, rbf kernel, no standardization) was trained using 100 vocalizations, of which 50 had pronounced periodicities in the 1.1–2.5 kHz according to their TMS and 50 which did not have them. These vocalizations have been used for the same purpose in another study [25] and were picked randomly from the original dataset after visual inspection.

Normality was tested for distributions of continuous variables using the Kolmogorov–Smirnov test. Since they were not normally distributed, statistical comparisons were performed using two-sided Wilcoxon rank-sum tests at 5% significance level. For the sex comparisons of the calls' parameters, the vocalizations were treated as independent. Cliff's delta ($d$) [65], an effect size measure, is also reported as a complement to null-hypothesis testing of rank sum tests. Cliff's $d$'s magnitude can be assessed using these thresholds of the absolute value [66]: $d < 0.147$ 'negligible', $d < 0.33$ 'small', $d < 0.474$ 'medium', otherwise 'large'. Pearson's correlation coefficient was used to measure the association between two continuous variables. Chi-squared test was used between two categorical variables together with a measure of effect size. Cohens's $d$ was used when each of the categories has two levels [67]: small, 0.20; medium, 0.50 and large, 0.80. Cramér's V was used when at least one of the categories has more than two levels ([68]; for its interpretation, see [69]).

In order to classify the syllables, the spectrogram cross-correlation algorithm from Avisoft was used, allowing tolerance values of ± 1 kHz and ± 0.1 ms in the frequency and time domain, respectively. This method compares the spectrogram of each syllable to an assembly of spectrograms of STs. This classification process has two steps. First, the assembly of STs is empty, since it builds from syllables that are poorly correlated with the existing templates. Therefore, the first syllable considered was also the first ST. The threshold criterion for the identification of new STs was 0.5. Second, each of the syllables was compared to the entire ST assembly created in the former step. The syllables are considered to belong to the ST with which they have the highest correlation value. The spectrogram cross-correlation algorithm is very sensitive to small differences that are not perceived by visual inspection. There were in total 265 different STs. The spectrograms of these STs were classified into five distress categories (modified from [33]) plus echolocation by visual inspection.

Ethics. The experiments comply with all current German laws on animal experimentation (Regierungspräsidium Darmstadt, experimental permit # F104/57).

Data accessibility. Data used in the manuscript to analyse the vocalisations are freely accessible online from the g-node database: https://doi.org/10.12751/g-node.tgej1e.

Authors' contributions. E.G.P., L.L.J and J.C.H. conceived the study. E.G.P., L.L.J., F.G.R. and J.C.H. conducted the experiments. E.G.P. analysed the data. E.G.P. wrote the first draft of the manuscript. E.G.P., L.L.J., F.G.R., J.W., A.K. and J.C.H. revised the manuscript.

Competing interests. The authors declare we have no competing interests.

Funding. This study was funded by Deutsche Forschungsgemeinschaft grant no. (275755787).

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
