## [Peer Review File · Royal Society Open Science]

Review History

RSOS-201254.R0 (Original submission)

Review form: Reviewer 1

Is the manuscript scientifically sound in its present form?

Yes

Are the interpretations and conclusions justified by the results?

No

Is the language acceptable?

No

Do you have any ethical concerns with this paper?

No

Have you any concerns about statistical analyses in this paper?

No

Recommendation?

Accept with minor revision (please list in comments)

Comments to the Author(s)

This paper reports the production of distress calls in a single context (hand held capture) for male and female *Carollia perspicillata* bats. The main finding of the manuscript is that male bats produce significantly more distress vocalizations in this context and that these vocalizations are louder, harsher and of lower frequency in males. The data in the manuscript is generally sound, although I raise one query below about how the frequency differences are assessed and interpreted. The manuscript requires some rewriting of the title, introduction and discussion, as outlined below.

The title of the paper should be changed as it implies that multiple contexts of distress were surveyed, and herein the bat distress calls were only collected in one context – ie holding the bats. The title should also mention the species of bat since this is clearly not a generalist feature of all bats, given the conflicting data in prior studies.

Although the language is generally acceptable in the manuscript, the introduction requires some rewriting. For example, the first sentence “Vocal communication is determinant in many animal species as a means of interaction with other life beings.” is worded quite strangely and is therefore hard to understand. The second sentence “One example of it are the screams, a type of vocalization widespread across the animal kingdom, present in humans [1,2], non-human primates [3], lizards [4], crocodylians [5], frogs [6], rats [7], birds [8] and bats [9,10], among other species.” refers to “screams”, which is not standard terminology used across these species, so instead it should use “distress calls”. The introduction should also present a better cross section of the background on production of distress calls in bats and other mammals. A passing reference is made to the study by Hörmann et al, 2020, but these and related relevant results should be more thoroughly presented. Particularly since the introduction is a little unclear in saying that the study did not explore sex differences. Rather, Hörmann et al reported that there were no sex differences in the production of or response to distress calls in this species. The introduction does not need to minimize the findings of prior papers to present the papers findings as novel. The discussion should also compare the findings in this study where sex differences were found, to prior studies such as Hörmann et al 2020 where sex differences were not found. This may be explained by species differences, but it is a point worth discussing given that they are conflicting findings from bats in the same family.

The discussion would also needs some rewriting, since as written it is not particularly convincing. Although the text lists 5 hypotheses regarding distress calling, they are not all discussed with regards to the data and the prior research. Instead the text of the discussion seems biased towards the kin warning hypothesis. However, this seems unjustified based on the text and data as presented. Furthermore, the conclusions should be refined as they are made a little too strongly given the data provided. The text acknowledges caveats on lines 189-199, but the data presented here does little to support or refute any of the possible hypotheses regarding distress call function as it is purely concerned with distress call production in an artificial context and does not show any functional or ecological use or response to distress calls in this species. The discussion text should be very careful not to draw conclusions that are not supported by the data. Such overstating of the conclusions is exemplified in the last few sentences of the discussion: “To summarize, we have shown that there are sex differences in *C. perspicillata* distress calling behavior. Such differences could be linked to the social organization strategies of this species and might allow male bats to protect their kin. Our results suggest the existence of altruism in bats reflected in their calling behavior when under duress.” There is no evidence presented that the sex differences are linked to social organization or protection, and I cannot find

justification that these data suggest the existence of altruism in bats. As such the discussion should be carefully rewritten so that it does not overstate the results and so that the conclusions are supported by the data.

In Figure 2E, male and female frequency differences are compared and a significant difference is found. However I would like to query how these data were analyzed as it seems that the differences may be driven by a different call type produced by the females, not found in the males (fig 1B). If I interpret these data correctly, males and females both produce one type of non-fAM, and one type of fAM that look similar between sexes and have comparable frequency. Then there seems to be another type of fAM that is different between males and females. The distribution seen in Fig 2E is likely driven by this different fAM call that is high frequency and only found in females. It seems that when they employ the same call types, there is no difference in frequency, and all frequency differences are driven by this female specific call type. It would be worthwhile to split these call types an assess how and when these different call types are produced in males and females, compared to the shared call types, and how frequency varies across sexes in the shared vs different call types.

Review form: Reviewer 2

Is the manuscript scientifically sound in its present form?

No

Are the interpretations and conclusions justified by the results?

No

Is the language acceptable?

Yes

Do you have any ethical concerns with this paper?

No

Have you any concerns about statistical analyses in this paper?

No

Recommendation?

Major revision is needed (please make suggestions in comments)

Comments to the Author(s)

See attached file (Appendix A).

Decision letter (RSOS-201254.R0)

Dear Ms González-Palomares,

The Editors assigned to your paper RSOS-201254 "Male bats call more than females in distressful contexts" have made a decision based on their reading of the paper and any comments received from reviewers.

Regrettably, in view of the reports received, the manuscript has been rejected in its current form. However, a new manuscript may be submitted which takes into consideration these comments.

We invite you to respond to the comments supplied below and prepare a resubmission of your manuscript. Below the referees' and Editors' comments (where applicable) we provide additional requirements. We provide guidance below to help you prepare your revision.

Please note that resubmitting your manuscript does not guarantee eventual acceptance, and we do not generally allow multiple rounds of revision and resubmission, so we urge you to make every effort to fully address all of the comments at this stage. If deemed necessary by the Editors, your manuscript will be sent back to one or more of the original reviewers for assessment. If the original reviewers are not available, we may invite new reviewers.

Please resubmit your revised manuscript and required files (see below) no later than 09-Mar-2021. Note: the ScholarOne system will 'lock' if resubmission is attempted on or after this deadline. If you do not think you will be able to meet this deadline, please contact the editorial office immediately.

Please note article processing charges apply to papers accepted for publication in Royal Society Open Science (<https://royalsocietypublishing.org/rsos/charges>). Charges will also apply to papers transferred to the journal from other Royal Society Publishing journals, as well as papers submitted as part of our collaboration with the Royal Society of Chemistry (<https://royalsocietypublishing.org/rsos/chemistry>). Fee waivers are available but must be requested when you submit your manuscript (<https://royalsocietypublishing.org/rsos/waivers>).

Thank you for submitting your manuscript to Royal Society Open Science and we look forward to receiving your resubmission. If you have any questions at all, please do not hesitate to get in touch.

Best regards,

on behalf of Dr Claudia Wascher (Associate Editor) and Kevin Padian (Subject Editor)
openscience@royalsociety.org

Subject Editor Comments to Author:

Thanks for your submission. As you can see some concerns were raised that suggest a reformulation of some of the research, which hopefully will be possible. If you choose to resubmit, please attend to the individual comments of the reviewers and AE. Best wishes.

Associate Editor Comments to Author (Dr Claudia Wascher):

Thank you for submitting this paper for consideration to Royal Society Open Science. We have received comments from two reviewers, and they both find the presented study technically

sound. Concerns arise whether the results actually support the conclusion and whether the hypothesis was fitted a posteriori and that the hypothesis is not well supported by the described biology of the focal species. I also find the description of the methods is lacking a lot of detail, for example how long were the bats kept in captivity, detailed description of the handling procedure (how long did it take at all). In my opinion, a crucial control condition is missing, for example are male individuals generally calling more. Could the results of the experiment be explained by personality differences, for example male individuals being bolder and therefore calling more? I do not think that the authors have actually tested whether calling behaviour can be explained by altruism, which they claim in their conclusion.

Reviewer comments to Author:

Reviewer: 1

Comments to the Author(s)

This paper reports the production of distress calls in a single context (hand held capture) for male and female *Carollia perspicillata* bats. The main finding of the manuscript is that male bats produce significantly more distress vocalizations in this context and that these vocalizations are louder, harsher and of lower frequency in males. The data in the manuscript is generally sound, although I raise one query below about how the frequency differences are assessed and interpreted. The manuscript requires some rewriting of the title, introduction and discussion, as outlined below.

The title of the paper should be changed as it implies that multiple contexts of distress were surveyed, and herein the bat distress calls were only collected in one context – ie holding the bats. The title should also mention the species of bat since this is clearly not a generalist feature of all bats, given the conflicting data in prior studies.

Although the language is generally acceptable in the manuscript, the introduction requires some rewriting. For example, the first sentence “Vocal communication is determinant in many animal species as a means of interaction with other life beings.” is worded quite strangely and is therefore hard to understand. The second sentence “One example of it are the screams, a type of vocalization widespread across the animal kingdom, present in humans [1,2], non-human primates [3], lizards [4], crocodilians [5], frogs [6], rats [7], birds [8] and bats [9,10], among other species.” refers to “screams”, which is not standard terminology used across these species, so instead it should use “distress calls”. The introduction should also present a better cross section of the background on production of distress calls in bats and other mammals. A passing reference is made to the study by Hörmann et al, 2020, but these and related relevant results should be more thoroughly presented. Particularly since the introduction is a little unclear in saying that the study did not explore sex differences. Rather, Hörmann et al reported that there were no sex differences in the production of or response to distress calls in this species. The introduction does not need to minimize the findings of prior papers to present the papers findings as novel. The discussion should also compare the findings in this study where sex differences were found, to prior studies such as Hörmann et al 2020 where sex differences were not found. This may be explained by species differences, but it is a point worth discussing given that they are conflicting findings from bats in the same family.

The discussion would also needs some rewriting, since as written it is not particularly convincing. Although the text lists 5 hypotheses regarding distress calling, they are not all discussed with regards to the data and the prior research. Instead the text of the discussion seems biased towards the kin warning hypothesis. However, this seems unjustified based on the text and data as presented. Furthermore, the conclusions should be refined as they are made a little too strongly given the data provided. The text acknowledges caveats on lines 189-199, but the

data presented here does little to support or refute any of the possible hypotheses regarding distress call function as it is purely concerned with distress call production in an artificial context and does not show any functional or ecological use or response to distress calls in this species. The discussion text should be very careful not to draw conclusions that are not supported by the data. Such overstating of the conclusions is exemplified in the last few sentences of the discussion: "To summarize, we have shown that there are sex differences in *C. perspicillata* distress calling behavior. Such differences could be linked to the social organization strategies of this species and might allow male bats to protect their kin. Our results suggest the existence of altruism in bats reflected in their calling behavior when under duress." There is no evidence presented that the sex differences are linked to social organization or protection, and I cannot find justification that these data suggest the existence of altruism in bats. As such the discussion should be carefully rewritten so that it does not overstate the results and so that the conclusions are supported by the data.

In Figure 2E, male and female frequency differences are compared and a significant difference is found. However I would like to query how these data were analyzed as it seems that the differences may be driven by a different call type produced by the females, not found in the males (fig 1B). If I interpret these data correctly, males and females both produce one type of non-fAM, and one type of fAM that look similar between sexes and have comparable frequency. Then there seems to be another type of fAM that is different between males and females. The distribution seen in Fig 2E is likely driven by this different fAM call that is high frequency and only found in females. It seems that when they employ the same call types, there is no difference in frequency, and all frequency differences are driven by this female specific call type. It would be worthwhile to split these call types and assess how and when these different call types are produced in males and females, compared to the shared call types, and how frequency varies across sexes in the shared vs different call types.

Reviewer: 2

Comments to the Author(s)

See attached file.

===PREPARING YOUR MANUSCRIPT===

While not essential, it will speed up the preparation of your manuscript proof if accepted if you format your references/bibliography in Vancouver style (please see

<https://royalsociety.org/journals/authors/author-guidelines/#formatting>). You should include DOIs for as many of the references as possible.

===PREPARING YOUR REVISION IN SCHOLARONE===

Author's Response to Decision Letter for (RSOS-201254.R0)

See Appendix B.

RSOS-202336.R0

Review form: Reviewer 1

Is the manuscript scientifically sound in its present form?

Yes

Are the interpretations and conclusions justified by the results?

Yes

Is the language acceptable?

Yes

Do you have any ethical concerns with this paper?

No

Have you any concerns about statistical analyses in this paper?

No

Recommendation?

Accept with minor revision (please list in comments)

Comments to the Author(s)

The authors have addressed most of the issues raised in the initial submission. The extra analysis of call types is a valuable addition. There are a few small outstanding issues that should be addressed prior to acceptance:

1. The title was changed to: "Male *Carollia perspicillata* bats call more than females in distressful contexts" to include the species name, which is helpful. However, it does not address the criticism that there are not multiple "contexts" investigated here, only a single hand-held

recording context. A sentence in the discussion about the potential behavioural differences in other contexts does not warrant a plural in the title – it is misleading about the scope of the data collected as it is. Therefore, the title “Male *Carollia perspicillata* bats call more than females in a distressful context” would be more accurate and something along these lines should be used instead.

2. I still find “life beings” a very strange way to say living beings or living animals, but I leave that to the authors and copy editors to decide.
3. The statement on line 17-18 should be qualified to indicate that “...in *C. perspicillata* bats, males are more likely to produce distress vocalizations than females.” is in the context tested. A blanked statement cannot be made without qualification given the studies herein.
4. The discussion of oxytocin and vasopressin is now perhaps a little long given its tangential nature for the results. I recommend a slight shortening of this part of the discussion since I do not think so much detail is necessary.

Decision letter (RSOS-202336.R0)

Dear Ms González-Palomares

On behalf of the Editors, we are pleased to inform you that your Manuscript RSOS-202336 "Male *Carollia perspicillata* bats call more than females in distressful contexts" has been accepted for publication in Royal Society Open Science subject to minor revision in accordance with the referees' reports. Please find the referees' comments along with any feedback from the Editors below my signature.

Please submit your revised manuscript and required files (see below) no later than 7 days from today's (ie 12-Apr-2021) date. Note: the ScholarOne system will 'lock' if submission of the revision is attempted 7 or more days after the deadline. If you do not think you will be able to meet this deadline please contact the editorial office immediately.

Kind regards,
Royal Society Open Science Editorial Office
Royal Society Open Science

on behalf of Dr Claudia Wascher (Associate Editor) and Kevin Padian (Subject Editor)
openscience@royalsociety.org

Associate Editor Comments to Author (Dr Claudia Wascher):

Comments to the Author:

Apologies for the delay in coming back to you. A second reviewer had agreed to review your manuscript, but did not return a report. One reviewer provided a detailed report and recommends accept with minor revisions, which I agree and recommend.

Reviewer comments to Author:

Reviewer: 1

Comments to the Author(s)

The authors have addressed most of the issues raised in the initial submission. The extra analysis of call types is a valuable addition. There are a few small outstanding issues that should be addressed prior to acceptance:

1. The title was changed to: "Male *Carollia perspicillata* bats call more than females in distressful contexts" to include the species name, which is helpful. However, it does not address the criticism that there are not multiple "contexts" investigated here, only a single hand-held recording context. A sentence in the discussion about the potential behavioural differences in other contexts does not warrant a plural in the title – it is misleading about the scope of the data collected as it is. Therefore, the title "Male *Carollia perspicillata* bats call more than females in a distressful context" would be more accurate and something along these lines should be used instead.
2. I still find "life beings" a very strange way to say living beings or living animals, but I leave that to the authors and copy editors to decide.
3. The statement on line 17-18 should be qualified to indicate that "...in *C. perspicillata* bats, males are more likely to produce distress vocalizations than females." is in the context tested. A blanked statement cannot be made without qualification given the studies herein.
4. The discussion of oxytocin and vasopressin is now perhaps a little long given its tangential nature for the results. I recommend a slight shortening of this part of the discussion since I do not think so much detail is necessary.

===PREPARING YOUR MANUSCRIPT===

===PREPARING YOUR REVISION IN SCHOLARONE===

<https://royalsociety.org/journals/authors/author-guidelines/#data>. You should ensure that

you cite the dataset in your reference list. If you have deposited data etc in the Dryad repository, please only include the 'For publication' link at this stage. You should remove the 'For review' link.

Author's Response to Decision Letter for (RSOS-202336.R0)

See Appendix C.

Decision letter (RSOS-202336.R1)

Dear Ms González-Palomares,

I am pleased to inform you that your manuscript entitled "Male *Carollia perspicillata* bats call more than females in a distressful context" is now accepted for publication in Royal Society Open Science.

Please see the Royal Society Publishing guidance on how you may share your accepted author manuscript at <https://royalsociety.org/journals/ethics-policies/media-embargo/>. After

publication, some additional ways to effectively promote your article can also be found here <https://royalsociety.org/blog/2020/07/promoting-your-latest-paper-and-tracking-your-results/>.

on behalf of Dr Claudia Wascher (Associate Editor) and Kevin Padian (Subject Editor)
openscience@royalsociety.org

Appendix A

General comments:

The manuscript investigates sex-specific differences in distress calling behavior in captive bats, *Carollia perspicillata*. This topic has received little attention so far and the authors collected high quality acoustic recordings which they analysed with state-of-the-art methods.

However, the whole manuscript reads as if the main hypothesis - males produce more distress calls than females - is an a posteriori construction based on the outcome of the analysis since the authors make no attempt to provide data for the rationale of the hypothesis (males profit more than females from warning kin).

In my opinion, the manuscript has two major weak points:

1) The lack of any behavioral data which would be needed to strengthen the authors main hypothesis (male *C. perspicillata* produce more distress calls than females because they profit most from warning kin).

This hypothesis is not well supported by what is known about the species' natural history (see specific comments below) Several other explanations could explain the observed difference in calling behavior (e.g. testosterone-based aggressiveness, sex-specific personality differences, etc.) and should be discussed accordingly. Without any behavioral data (e.g. males produce more distress calls near the roost than females or than other males in the foraging grounds; males produce more distress calls when they currently have offspring in the colony; etc.) this hypothesis is difficult to uphold.

2) The narrow and superfluous framework of the introduction and discussion.

Both sections appear to be surprisingly uncaring about the subject at hand. They should be thoroughly rewritten and expanded to adequately convey the depth and importance of the topic (sex-specific differences in calling behavior).

Also, I would like to make two points here: a) Mixing arguments about alarm calls and distress calls should be generally avoided as these call types may have very different functions and receivers. b) In the discussion, the section about altruism seems far fetched to me. While warning kin may be altruistic (or not, depending on whom you ask), it is completely unclear if male *C. perspicillata* actually produce distress calls to warn kin. Until this is proven, it does not make much sense to speculate about altruism.

Specific comments:

Line 34: Indicate is a strong word. I suggest to rephrase the sentence in a more cautious way. Males could call more because they are more aggressive than females or bolder or....

"reaching other individuals" is just one of many explanations for your findings.

Lines 36-37: This statement ("males would be the sex with the highest number of relatives among the receivers") is only correct if males are philopatric and/or sire most offspring in their harem. Both aspects are not well documented for *C. perspicillata*. According to Fleming (1988), offspring dispersal from the natal colony seems to be slightly female biased but pups of both sexes may also remain in their natal colony (in which case they only leave their natal harem territory). Porter and McCracken (1983) report that even though harem males have priority access to females roosting in their territories, both bachelor and other harem males attempt to copulate with estrus females as well.

Lines 62-72: I find these examples to be off-topic, as they cover alarm calls and not distress calls, aka screams. Please select more appropriate examples.

Line 73: I suggest citing Nagel (2006), since it provides a multi-species comparison on distress calling in bats. Table 2 is especially relevant (see below).

Nagel J (2006) Variation in distress calls of New World bats. M.Sc. Thesis. Faculty of Graduate Studies, University of Western Ontario, Canada.

Table 2. Percent of individuals that produced distress calls while held in the hand and touched on the back and neck, by family and species. *Myotis lucifugus* were captured in Ontario, Canada. All others were captured in Belize. Chi-square test examines whether males or females were more likely to call. NA indicates the Chi-square test was not applicable (expected counts less than 5, or results were constant). * indicates significant results of $p < 0.05$.

	Total caught	% that called total	Males caught	% males that called	Females caught	% females that called	χ^2	df	p
Family Emballonuridae (Saccopteryx bilineata)	11	100.0%	3	100.0%	8	100.0%	NA	NA	NA
Family Mormoopidae (Pteronotus parnellii)	34	50.0%	2	100.0%	32	46.9%	NA	NA	NA
Family Vespertilionidae (2 species)	23	91.3%	3	100.0%	20	90.0%	NA	NA	NA
Bauerus dubiaquercus	12	100.0%	2	100.0%	10	100.0%	NA	NA	NA
Myotis lucifugus	11	86.1%	1	100.0%	10	80.0%	NA	NA	NA
Family Phyllostomidae (7 species)	325	43.7%	174	54.6%	151	31.1%	17.499	1	< 0.001*
Desmodus rotundus	22	100.0%	18	100.0%	4	100.0%	NA	NA	NA
Carollia brevicauda	36	86.1%	20	95.0%	16	75.0%	NA	NA	NA
Glossophaga soricina	58	63.8%	25	72.0%	33	57.6%	1.281	1	0.258
Artibeus lituratus	22	50.0%	17	47.1%	5	60.0%	NA	NA	NA
Artibeus intermedius	24	45.8%	18	55.6%	6	16.7%	NA	NA	NA
Artibeus jamaicensis	53	18.9%	25	28.0%	28	10.7%	2.578	1	0.108
Sturnira lilium	110	18.2%	51	29.4%	59	8.5%	8.061	1	0.005*
Total (11 species)	393	47.6%	181	56.4%	201	39.8%	10.461	1	0.001*

Lines 90-93, 149-151 and 191-193: I find this argument to be rather far-fetched. First, females invest much more in offspring than males (and have knowledge about paternity!) so it would be equally likely that they produce more distress calls (see also comment above). Second, it is not clear that distress calls are produced to warn kin. They could be used to startle a predator or request help. Third, distress calls are produced not only at the roost but also at the foraging sites.

Line 110: I do not understand why you cite these two studies here. Neither of them states that males utter more distress calls than females.

Figure 1: Please also show a spectrogram (not only an oscillogram) of a distress call series to see how they compare to previously published distress calls.

Line 165-169: This paragraph is superfluous. As stated before, alarm calls and distress calls are not the same and the reason for producing them may be very different. Also, neither social complexity nor hierarchies are discussed in [20] or in the introduction.

Lines 170-177: A relevant paper to discuss here would be Eckenweber & Knörnschild (2016). It shows that distress calls within or in proximity to the day-roost have a higher behavioural relevance than distress calls at foraging sites. Also, it shows that both male and female *S. bilineata* are equally likely to produce distress calls (in accordance with Table 2 in Nagel 2006).

Lines 178-188: This section is highly speculative. While warning kin may be altruistic, it is completely unclear if male *C. perspicillata* actually produce distress calls to warn kin. Until this is proven, it does not make much sense to speculate about altruism.

Lines 194-196: Please elaborate what you mean with this.

Appendix B

To: Editorial Board Member
Royal Society Open Science

23rd December 2020

Dear Editor:

We would like to resubmit the manuscript RSOS-201254 now titled “Male *Carollia perspicillata* bats call more than females in distressful contexts” for your consideration as research article in *Royal Society Open Science*.

We are thankful to the editor and both anonymous reviewers for having taken the time to review our manuscript. Both reviewers proposed a major change in the introduction and discussion sections. We have modified the ms following their questions and comments. The result is a much-improved paper that we believe will make a meaningful contribution to the current body-of-knowledge on animal vocal behaviour. Major modifications to the paper include:

- The addition of a vocalization classification based on the spectrogram cross-correlation algorithm in order to look for sex-specific vocalizations as suggested by reviewer 1 (in Figs 3 and 4).
- Rewriting of most of the introduction as suggested by both reviewers. The citations have been revised and now it provides a broader picture on distress calling.
- Complete rewriting of the discussion as indicated by both reviewers, with less focus on trying to find a potential function in distress calling, but rather mentioning literature that has observed different patterns in vocal behaviour and their correlation with e.g. signalling and hormonal systems.

A more detailed response to the specific comments can be found below. At the end of the document there is the reference list of the articles mentioned in our responses. Thanks for giving us the chance to resubmit our article. We look forward to your comments.

As corresponding authors, we confirm that the manuscript has been read and approved for submission by all the named authors. We hope you find our resubmitted manuscript suitable for publication and look forward to hearing from you in due course.

Yours sincerely,

The corresponding authors

Comments and responses (editor and reviewers' comments in black and our answers in blue):

Associate Editor Comments to Author (Dr Claudia Wascher):

Thank you for submitting this paper for consideration to Royal Society Open Science. We have received comments from two reviewers, and they both find the presented study technically sound. Concerns arise whether the results actually support the conclusion and whether the hypothesis was fitted a posteriori and that the hypothesis is not well supported by the described biology of the focal species. I also find the description of the methods is lacking a lot of detail, for example how long were the bats kept in captivity, detailed description of the handling procedure (how long did it take at all). In my opinion, a crucial control condition is missing, for example are male individuals generally calling more. Could the results of the experiment be explained by personality differences, for example male individuals being bolder and therefore calling more? I do not think that the authors have actually tested whether calling behaviour can be explained by altruism, which they claim in their conclusion.

We thank Dr Claudia Wascher, Associate Editor, for her time reading and evaluating our manuscript. We have now modified the introduction and discussion, where we discuss possible factors that could influence the sex-specific, or even intrasexual differences. We don't try to suggest a potential function for distress calling, since we lack relevant behavioural data. We hope the Associate Editor will find the current version more suited for publication.

Regarding the methods. Bats come from a colony located in the facilities of our institute, therefore, they have always been in captivity. The total duration of the recording session was 3 min. This information is found in:

Lines 288-289: "The bats were taken in small groups from the colony and brought to a sound-proof booth for the individual vocalizations recording (3-min long)..."

Lines 282-284: "The bats were taken from the bat colony at the Institute for Cell Biology and Neuroscience at the Goethe University in Frankfurt am Main, Germany."

Reviewer: 1

Comments to the Author(s)

This paper reports the production of distress calls in a single context (hand-held capture) for

male and female *Carollia perspicillata* bats. The main finding of the manuscript is that male bats produce significantly more distress vocalizations in this context and that these vocalizations are louder, harsher and of lower frequency in males. The data in the manuscript is generally sound, although I raise one query below about how the frequency differences are assessed and interpreted. The manuscript requires some rewriting of the title, introduction and discussion, as outlined below.

1. The title of the paper should be changed as it implies that multiple contexts of distress were surveyed, and herein the bat distress calls were only collected in one context – ie holding the bats. The title should also mention the species of bat since this is clearly not a generalist feature of all bats, given the conflicting data in prior studies.

Thank you for the comment. The title includes now the bat species (“Male *Carollia perspicillata* bats call more than females in distressful contexts”) and a sentence in the discussion mentions the potential behavioural difference in other distressful contexts.

Lines 252-254: “It is worth noting also that other distressful contexts, such as captured by a natural predator, could have different results than what has been reported here.”

2. Although the language is generally acceptable in the manuscript, the introduction requires some rewriting. For example, the first sentence “Vocal communication is determinant in many animal species as a means of interaction with other life beings.” is worded quite strangely and is therefore hard to understand.

Thank you for pointing this out. We revised this version of the paper to check for grammatical errors. The sentence mentioned by the reviewer now reads: (line 34): “Vocal communication is essential as a means of interaction among life beings”.

3. The second sentence “One example of it are the screams, a type of vocalization widespread across the animal kingdom, present in humans [1,2], non-human primates [3], lizards [4], crocodylians [5], frogs [6], rats [7], birds [8] and bats [9,10], among other species.” refers to “screams”, which is not standard terminology used across these species, so instead it should use “distress calls”.

We agree that the “screams” term is not completely interchangeable with “distress calls”.

Lines 34-35: “One example of it are the distress calls, a type of vocalization widespread across the animal kingdom,” ...

3. The introduction should also present a better cross section of the background on production of distress calls in bats and other mammals. A passing reference is made to the study by Hörmann et al, 2020, but these and related relevant results should be more thoroughly presented. Particularly since the introduction is a little unclear in saying that the study did not explore sex differences. Rather, Hörmann et al reported that there were no sex differences in the production of or response to distress calls in this species. The introduction does not need to minimize the findings of prior papers to present the papers findings as novel. The discussion should also compare the findings in this study where sex differences were found, to prior studies such as Hörmann et al 2020 where sex differences were not found. This may be explained by species differences, but it is a point worth discussing given that they are conflicting findings from bats in the same family.

We are sorry for not having stated it clear in the text. Our intention was, by no means, to minimize was done in previous studies. In the current version of the manuscript, the study is mentioned in lines 72-78:

“In bats, previous studies had shown sex differences in distress calling behaviour in species such as *Sturnira lilium* in which males were more likely to produce distress sounds than females (Carter et al., 2015). However, in other bat species (*Glossophaga soricina*, (Hörmann et al., 2020), *Rhinolophus ferrumequinum* (Jiang et al., 2017)) this sex distinction in distress calling was not as clear. Although these studies have alluded to sex differences -or its nonexistence- in bat distress calling behaviour, in-depth information about sex differences in distress calling in bats, i.e. how acoustic parameters of the calls change with sex, is presently lacking.”

And in the discussion again (lines 181-187):

“Previous studies in bats on sex differences in the vocal behaviour are scarce. Carter et al. 2015 reported that male bats of the species *Sturnira lilium* are more likely to produce distress sounds than females. On the contrary, a large proportion of females and males of the species *Glossophaga soricina* emitted distress calls (65% and 50%, respectively) when handled, but no significant differences were found (Hörmann et al., 2020). Male and female *Rhinolophus*

ferrumequinum do not show a sex-specific behaviour regarding distress calling in parameters like call rate, fundamental frequency or duration (Jiang et al., 2017).”

4. The discussion would also need some rewriting, since as written it is not particularly convincing. Although the text lists 5 hypotheses regarding distress calling, they are not all discussed with regards to the data and the prior research. Instead the text of the discussion seems biased towards the kin warning hypothesis. However, this seems unjustified based on the text and data as presented. Furthermore, the conclusions should be refined as they are made a little too strongly given the data provided. The text acknowledges caveats on lines 189-199, but the data presented here does little to support or refute any of the possible hypotheses regarding distress call function as it is purely concerned with distress call production in an artificial context and does not show any functional or ecological use or response to distress calls in this species. The discussion text should be very careful not to draw conclusions that are not supported by the data. Such overstating of the conclusions is exemplified in the last few sentences of the discussion: “To summarize, we have shown that there are sex differences in *C. perspicillata* distress calling behavior. Such differences could be linked to the social organization strategies of this species and might allow male bats to protect their kin. Our results suggest the existence of altruism in bats reflected in their calling behavior when under duress.” There is no evidence presented that the sex differences are linked to social organization or protection, and I cannot find justification that these data suggest the existence of altruism in bats. As such the discussion should be carefully rewritten so that it does not overstate the results and so that the conclusions are supported by the data.

We thank the reviewer for these insightful comments. The whole “Discussion” section has been rewritten accordingly. Our intentions were never to provide a reason for the different behaviour in male and females, rather a hypothesis/suggestion. Since, as stated by both reviewers, we do not show any behavioural data to corroborate our speculations, in the rewriting we have not speculated about potential functions of distress calling, but rather to suggest based on the literature possible signalling pathways, hormonal effects, environmental factors that could be at the source of the distinct behaviour of male bats described in our paper.

5. In Figure 2E, male and female frequency differences are compared and a significant difference is found. However I would like to query how these data were analyzed as it seems

that the differences may be driven by a different call type produced by the females, not found in the males (fig 1B). If I interpret these data correctly, males and females both produce one type of non-fAM, and one type of fAM that look similar between sexes and have comparable frequency. Then there seems to be another type of fAM that is different between males and females. The distribution seen in Fig 2E is likely driven by this different fAM call that is high frequency and only found in females. It seems that when they employ the same call types, there is no difference in frequency, and all frequency differences are driven by this female specific call type. It would be worthwhile to split these call types and assess how and when these different call types are produced in males and females, compared to the shared call types, and how frequency varies across sexes in the shared vs different call types.

Thank you for this observation. We have re-visited the data searching for syllable types that might be specific to females and males. The syllable classification algorithm was the spectral cross-correlation algorithm implemented in Avisoft. This resulted in a number of 265 syllable types (different designs). To reduce this number, and following Kanwal et. al. suggestions (1994, JASA), the templates of these syllable types were classified visually into “echolocation calls” and five distress calls categories: downward frequency modulated sounds (dFM), sinusoidal frequency modulated (sFM), noise burst (NB), sinusoidal noise burst (sNB) and quasi constant frequency (qCF). The results of these analyses are in Figs. 3 and 4.

Briefly, we observed that in all distress categories except for qCF and echolocation calls, there was a sex difference between the area under the curve in the low and high-frequency ranges (10-40 kHz and 45-90 kHz, respectively). In addition, there were sex differences in the fAM/non-fAM proportion in the same categories, i.e. dFM, sFM, NB and sNB. These results can be found in the new ms version (pp. 5-6).

Reviewer: 2

Comments to the Author(s)

General comments:

The manuscript investigates sex-specific differences in distress calling behavior in captive bats, *Carollia perspicillata*. This topic has received little attention so far and the authors collected high quality acoustic recordings which they analysed with state-of-the-art methods.

However, the whole manuscript reads as if the main hypothesis -males produce more distress calls than females- is an a posteriori construction based on the outcome of the analysis since the authors make no attempt to provide data for the rationale of the hypothesis (males profit more than females from warning kin). In my opinion, the manuscript has two major weak points:

1) The lack of any behavioral data which would be needed to strengthen the authors main hypothesis (male *C. perspicillata* produce more distress calls than females because they profit most from warning kin). This hypothesis is not well supported by what is known about the species' natural history(see specific comments below)Several other explanations could explain the observed difference in calling behavior (e.g. testosterone-based aggressiveness, sex-specific personality differences, etc.) and should be discussed accordingly. Without any behavioral data (e.g. males produce more distress calls near the roost than females or than other males in the foraging grounds; males produce more distress calls when they currently have offspring in the colony; etc.) this hypothesis is difficult to uphold.

We thank the reviewer for this criticism, and we completely agree on that. The introduction and discussion have been rewritten accordingly.

2) The narrow and superfluous framework of the introduction and discussion. Both sections appear to be surprisingly uncaring about the subject at hand. They should be thoroughly rewritten and expanded to adequately convey the depth and importance of the topic (sex-specific differences in calling behavior). Also, I would like to make two points here: a) Mixing arguments about alarm calls and distress calls should be generally avoided as these call types may have very different functions and receivers. b) In the discussion, the section about altruism seems far-fetched to me. While warning kin may be altruistic (or not, depending on whom you ask), it is completely unclear if male *C. perspicillata* actually produce distress calls to warn kin. Until this is proven, it does not make much sense to speculate about altruism.

We appreciate the comment, but we should mention that the way this constructive criticism was formulated is -in our opinion- unnecessarily harsh.

We have rewritten the introduction and discussion accordingly. The current ms version does not consider examples of alarm calling and we do not focus on the kin warning hypothesis as the potential function of distress calling. Instead the introduction is more general, cites some

examples of distress calling and the research on sex differences in distress calling in bats. The new discussion now does not try to provide a possible behavioural reason for the sex different behaviour in distress calling. Instead it lists a series of hormonal, neurotransmitter, and behavioural differences that may be behind our reported results.

Specific comments:

Line 34: Indicate is a strong word. I suggest to rephrase the sentence in a more cautious way.

Males could call more because they are more aggressive than females or bolder or....

“reaching other individuals” is just one of many explanations for your findings.

Thank you for this suggestion. This part of the abstract is not present anymore in the new version.

Lines 36-37: This statement (“males would be the sex with the highest number of relatives among the receivers”) is only correct if males are philopatric and/or sire most offspring in their harem. Both aspects are not well documented for *C. perspicillata*. According to Fleming (1988), offspring dispersal from the natal colony seems to be slightly female biased but pups of both sexes may also remain in their natal colony (in which case they only leave their natal harem territory). Porter and McCracken (1983) report that even though harem males have priority access to females roosting in their territories, both bachelor and other harem males attempt to copulate with estrus females as well.

We thank the reviewer for this clarification. These arguments are not present in the new manuscript.

Lines 62-72: I find these examples to be off-topic, as they cover alarm calls and not distress calls, aka screams. Please select more appropriate examples.

Thank you for pointing this out, and we apologize for this. We agree that these are different vocalization types and that the motivation for their utterance is different than for distress calls. These examples are not longer part of the revised manuscript.

Line 73: I suggest citing Nagel (2006), since it provides a multi-species comparison on distress calling in bats. Table 2 is especially relevant (see below).

Table 2. Percent of individuals that produced distress calls while held in the hand and touched on the back and neck, by family and species. *Myotis lucifugus* were captured in Ontario, Canada. All others were captured in Belize. Chi-square test examines whether males or females were more likely to call. NA indicates the Chi-square test was not applicable (expected counts less than 5, or results were constant). * indicates significant results of $p < 0.05$.

	Total caught	% that called total	Males caught	% males that called	Females caught	% females that called	χ^2	df	p
Family Emballonuridae (Saccopteryx bilineata)	11	100.0%	3	100.0%	8	100.0%	NA	NA	NA
Family Mormoopidae (Pteronotus parnellii)	34	50.0%	2	100.0%	32	46.9%	NA	NA	NA
Family Vespertilionidae (2 species)	23	91.3%	3	100.0%	20	90.0%	NA	NA	NA
Bauerus dubiaquercus	12	100.0%	2	100.0%	10	100.0%	NA	NA	NA
Myotis lucifugus	11	86.1%	1	100.0%	10	80.0%	NA	NA	NA
Family Phyllostomidae (7 species)	325	43.7%	174	54.6%	151	31.1%	17.499	1	< 0.001*
Desmodus rotundus	22	100.0%	18	100.0%	4	100.0%	NA	NA	NA
Carollia brevicauda	36	86.1%	20	95.0%	16	75.0%	NA	NA	NA
Glossophaga soricina	58	63.8%	25	72.0%	33	57.6%	1.281	1	0.258
Artibeus lituratus	22	50.0%	17	47.1%	5	60.0%	NA	NA	NA
Artibeus intermedius	24	45.8%	18	55.6%	6	16.7%	NA	NA	NA
Artibeus jamaicensis	53	18.9%	25	28.0%	28	10.7%	2.578	1	0.108
Sturnira lilium	110	18.2%	51	29.4%	59	8.5%	8.061	1	0.005*
Total (11 species)	393	47.6%	181	56.4%	201	39.8%	10.461	1	0.001*

Nagel J (2006) Variation in distress calls of New World bats. M.Sc. Thesis. Faculty of Graduate Studies, University of Western Ontario, Canada.

Thank you for this suggestion. We agree. The information in Nagel 2006 is very valuable. However, we would prefer to cite only peer-reviewed studies published in scientific journals. This decision is not only based on scientific quality but also on the availability of the studies now and in the far future. We hope the reviewer understands our reasoning.

Lines 90-93, 149-151 and 191-193: I find this argument to be rather far-fetched. First, females invest much more in offspring than males (and have knowledge about paternity!) so it would be equally likely that they produce more distress calls (see also comment above). Second, it is not clear that distress calls are produced to warn kin. They could be used to startle a predator or request help. Third, distress calls are produced not only at the roost but also at the foraging sites.

We appreciate the comment, and we agree that we do not have the relevant behavioural data to make those claims. These parts are not in the manuscript anymore.

Line 110: I do not understand why you cite these two studies here. Neither of them states that males utter more distress calls than females.

This part has been removed.

Figure 1: Please also show a spectrogram (not only an oscillogram) of a distress call series to see how they compare to previously published distress calls.

Thank you for the suggestion. We believe that spectrograms with such a large x-axis limits (180 s) are not satisfying their purpose in a paper-sized figure, since the spectrogram of each of the vocalizations would barely be distinguishable. Nonetheless, since we have performed an extra analysis on the syllable classification, we provide now more examples (oscillogram + spectrogram) in Fig. 3.

Line 165-169: This paragraph is superfluous. As stated before, alarm calls and distress calls are not the same and the reason for producing them may be very different. Also, neither social complexity nor hierarchies are discussed in [20] or in the introduction.

This part is no longer part in the manuscript.

Lines 170-177: A relevant paper to discuss here would be Eckenweber & Knörnschild (2016). It shows that distress calls within or in proximity to the day-roost have a higher behavioural relevance than distress calls at foraging sites. Also, it shows that both male and female *S. bilineata* are equally likely to produce distress calls (in accordance with Table 2 in Nagel 2006).

We appreciate the suggestion. The study is now mentioned in the introduction (lines 56-58): “It has also been shown that bats are able to modulate their responsiveness to distress calls depending on how distant from the day-roost they are emitted (Eckenweber and Knörnschild, 2016).”.

Lines 178-188: This section is highly speculative. While warning kin may be altruistic, it is completely unclear if male *C. perspicillata* actually produce distress calls to warn kin. Until this is proven, it does not make much sense to speculate about altruism.

This whole section has been rewritten.

Lines 194-196: Please elaborate what you mean with this.

We have removed this sentence.

Appendix C

To: Editorial Board Member
Royal Society Open Science

13th April 2021

We are very pleased that the manuscript has been accepted for publication in Royal Society Open Science and we are also thankful to the reviewer for taking the time to review our resubmission (manuscript RSOS-201254 now titled “Male *Carollia perspicillata* bats call more than females in a distressful context”). We have modified it according to the reviewer’s comments. A more detailed response to the specific comments is found below.

As corresponding authors, we confirm that the manuscript has been read and approved for submission by all the named authors. We hope you find our resubmitted manuscript suitable for publication and look forward to hearing from you in due course.

Yours sincerely,

The corresponding authors

Comments and responses (editor and reviewers’ comments in black and our answers in blue):

Associate Editor Comments to Author (Dr Claudia Wascher):

Comments to the Author:

Apologies for the delay in coming back to you. A second reviewer had agreed to review your manuscript but did not return a report. One reviewer provided a detailed report and recommends accept with minor revisions, which I agree and recommend.

Reviewer comments to Author:

Reviewer: 1

Comments to the Author(s)

The authors have addressed most of the issues raised in the initial submission. The extra analysis of call types is a valuable addition. There are a few small outstanding issues that should be addressed prior to acceptance:

1. The title was changed to: “Male *Carollia perspicillata* bats call more than females in distressful contexts” to include the species name, which is helpful. However, it does not address the criticism that there are not multiple “contexts” investigated here, only a single hand-held recording context. A sentence in the discussion about the potential behavioural differences in other contexts does not warrant a plural in the title – it is misleading about the scope of the data collected as it is. Therefore, the title “Male *Carollia perspicillata* bats call more than females in a distressful context” would be more accurate and something along these lines should be used instead.

Thank you for the suggestion. We have modified the title accordingly. As suggested, the current title is: “Male *Carollia perspicillata* bats call more than females in a distressful context”.

2. I still find “life beings” a very strange way to say living beings or living animals, but I leave that to the authors and copy editors to decide.

Changed to “living beings”.

3. The statement on line 17-18 should be qualified to indicate that “...in *C. perspicillata* bats, males are more likely to produce distress vocalizations than females.” is in the context tested. A blanked statement cannot be made without qualification given the studies herein.

Thank you for pointing that out. We have added that information and now the sentence reads (lines 17-18): “We show that in *C. perspicillata* bats, males are more likely to produce distress vocalizations than females when hand-held”.

4. The discussion of oxytocin and vasopressin is now perhaps a little long given its tangential nature for the results. I recommend a slight shortening of this part of the discussion since I do not think so much detail is necessary.

We appreciate the comment. The aforementioned paragraph (lines 221-228) has been simplified and now reads:

“Other hormonal systems that are sexually dimorphic, and which are associated with the mammalian reproductive and social behaviours are the oxytocin (OT) and arginine vasopressin (AVP) systems [49–51]. In non-human studies, these systems have been associated with sex-specific social behaviours [52], including vocal production. For example, intranasal administration of OT and AVP in female rhesus macaques resulted in an increase of aggression and “cooing” vocalizations towards males, but not females [49]. This evidence makes OT and AVP potential candidates for explaining the sex difference in distress calling reported here.”